# Evaluation of Statins Use in Hemodialysis Patients: A Retrospective Analysis of Clinical and Safety Outcomes

**DOI:** 10.3390/ph18060911

**Published:** 2025-06-18

**Authors:** Abdulmalik S. Alotaibi, Mohamed A. Albekery, Ahmed A. Alanazi, Ibrahim S. Alhomoud, Khalid A. Alamer, Mohammad Shawaqfeh, Reem H. Alshammari, Fayez Alhejaili, Muthana Al Sahlawi, Ibrahim Aldossary, Hajar Adel Aljuayl, Mohammad Alkathiri, Shmeylan Alharbi, Abdulkareem Albekairy, Abdulmalik Alkatheri

**Affiliations:** 1Pharmaceutical Practices Department, College of Pharmacy, Umm Al-Qura University, Makkah 21955, Saudi Arabia; 2Department of Pharmacy Practice, College of Clinical Pharmacy, King Faisal University, Al-Ahsa 31982, Saudi Arabia; malbekery@kfu.edu.sa (M.A.A.); hajer.adelaljuail@gmail.com (H.A.A.); 3Department of Pharmacy Practice, College of Pharmacy, Imam Abdulrahman Bin Faisal University, Dammam 31441, Saudi Arabia; aasalanazi@iau.edu.sa (A.A.A.); kaalamer@iau.edu.sa (K.A.A.); 4Department of Pharmacy Practice, College of Pharmacy, Qassim University, Qassim 51452, Saudi Arabia; i.alhomoud@qu.edu.sa; 5College of Pharmacy, King Saud Bin Abdulaziz University for Health Sciences, Riyadh 11481, Saudi Arabia; shawaqfehm@ksau-hs.edu.sa (M.S.); kathirim@ngha.med.sa (M.A.); harbishm@ngha.med.sa (S.A.); bekairya@ngha.med.sa (A.A.); katheria@mngha.med.sa (A.A.); 6King Abdullah International Medical Research Center, Ministry of National Guard Health Affairs, Riyadh 11426, Saudi Arabia; 7College of Pharmacy, Northern Border University, Arar 73213, Saudi Arabia; rehammad2@gmail.com; 8College of Medicine, King Saud Bin Abdulaziz University for Health Sciences, Riyadh 11481, Saudi Arabia; fhejaili@yahoo.com; 9Department of Internal Medicine, College of Medicine, King Faisal University, Hofuf 31982, Saudi Arabia; muthana.sahlawi@gmail.com; 10Pharmaceutical Care Services, King Abdulaziz Hospital, Ministry of National Guard Health Affairs, Hofuf 31982, Saudi Arabia; dossaryi2@mngha.med.sa; 11Pharmaceutical Care Department, King Abdulaziz Medical City, Riyadh 11426, Saudi Arabia

**Keywords:** end-stage renal diseases, hemodialysis, statins, dyslipidemia, efficacy, safety

## Abstract

**Background:** Lipid metabolism disturbances are common in end-stage renal disease (ESRD) patients on hemodialysis (HD), leading to dyslipidemia, which is characterized by abnormal plasma lipids and lipoproteins. Although large randomized controlled trials have generally not demonstrated a survival benefit associated with statin therapy among patients receiving hemodialysis, limited observational studies have reported potential associations with improved clinical outcomes in this population. **Methods**: This retrospective cohort study investigated the clinical and safety outcomes of statin use in ESRD patients on HD with documented dyslipidemia over a two-year period from 1 January 2018 to 30 December 2019. The primary endpoints evaluated the clinical outcomes of statins by assessing changes in specific lipid parameters, including low-density lipoprotein cholesterol (LDL-C), triglycerides (TG), total cholesterol (TC), and high-density lipoprotein cholesterol (HDL-C). The secondary endpoints assessed safety by monitoring liver enzymes and creatine kinase (CK) levels. **Results**: Among 179 participants, diabetes mellitus was present in 134 patients (74.9%), while 168 patients (93.9%) had hypertension. Cardiovascular events occurred in 95 patients (53.1%). Statin therapy was administered to 146 patients (82.0%), with atorvastatin being the most frequently prescribed statin (69.3%). Modest reductions in LDL-C levels were observed in the rosuvastatin and atorvastatin groups, whereas slight increases were noted in the simvastatin and non-statin groups. None of these within-group changes were statistically significant. In the atorvastatin group, LDL-C decreased slightly from 2.058 to 2.003 mmol/L. The rosuvastatin group experienced a more pronounced LDL-C reduction from 2.607 to 2.113 mmol/L. Conversely, the simvastatin group showed an LDL-C increase from 1.550 to 1.901 mmol/L. Among the non-statin group, LDL-C increased from 2.678 to 2.820 mmol/L. Liver enzyme and CK levels fluctuated slightly but remained within normal ranges. **Conclusions**: This study evaluated statin therapy in hemodialysis patients with dyslipidemia. Although modest reductions in LDL-C levels were observed in the atorvastatin and rosuvastatin groups, statin therapy did not reduce the incidence of atherosclerotic events in hemodialysis patients with dyslipidemia. Additionally, statin use was not associated with any clinically or statistically significant effects.

## 1. Introduction

Patients with end-stage renal disease (ESRD) are at a significantly higher risk of developing cardiovascular diseases (CVDs) such as coronary artery disease (CAD) and atherosclerotic coronary heart disease (ASCHD) compared to the general population. Cardiovascular event rates increase progressively as eGFR decreases [1,2,3]. The incidence rises from 2.11 events per 100 person-years at an eGFR ≥ 60 mL/min/1.73 m^2^ to 36.6 events per 100 person-years at an eGFR < 15 mL/min/1.73 m^2^ [1]. According to the United States Renal Data System (USRDS) 2022, cardiovascular diseases accounted for 40.1% of all-cause mortality in patients receiving hemodialysis (HD) and 51.5% in those undergoing peritoneal dialysis (PD) [4]. Arrhythmia and cardiac arrest were the leading causes of cardiovascular deaths in both groups [4]. Despite this high cardiovascular risk, statin therapy remains underused in patients with chronic kidney disease (CKD). An international investigation indicated that the utilization of statins among patients with CKD stages 3–5 varied by country, with rates between 49% and 60% [5].

Patients with ESRD on HD experience a significant burden of risk factors for CVD, including hypertension, diabetes mellitus, dyslipidemia, and other metabolic abnormalities [6]. A progressive decline in eGFR is associated with an increased prevalence of atherosclerotic lesions [7]. Therefore, several guidelines identify ESRD as a well-established risk factor for CVD [8,9,10]. Additionally, dyslipidemia is recognized as a significant mortality risk factor by the National Cholesterol Education Program Expert Panel [11]. ESRD patients exhibit a variety of plasma lipid and lipoprotein abnormalities, including elevated levels of oxidized low-density lipoprotein cholesterol (LDL-C), triglycerides (TG), and lipoprotein (a) [Lp(a)] [6,12]. Additional pathophysiological factors, such as arterial stiffness and calcification, significantly contribute to the risk of cardiovascular morbidity and mortality in patients with ESRD [7,13]. However, evidence regarding statin use in ESRD patients remains limited and inconclusive.

The 4D trial evaluated atorvastatin (20 mg daily) in 1255 HD patients with type 2 diabetes [14]. Despite achieving a 42% reduction in LDL-C after four weeks of atorvastatin treatment, no significant decrease was observed in the primary composite endpoint of cardiac death, nonfatal myocardial infarction, or stroke (HR 0.92; 95% CI, 0.77–1.10; *p* = 0.37). The AURORA trial results were also consistent with the 4D trial, in which rosuvastatin (10 mg daily) did not significantly reduce the primary composite endpoint of cardiovascular death, nonfatal myocardial infarction, or nonfatal stroke when compared with placebo (HR 0.96; 95% CI 0.84–1.11; *p* = 0.59). This finding highlights the complexity of lipid management in HD patients, as significant LDL-C reduction (43% at 3 months) did not translate into improved cardiovascular outcomes [15]. The SHARP trial expanded research in this field by including a larger cohort of 9270 patients with CKD, 3023 of whom were on dialysis. The study assessed the efficacy of simvastatin (20 mg daily) combined with ezetimibe (10 mg daily). Over a median follow-up period of 4.9 years, the intervention achieved a 17% reduction in the risk of major atherosclerotic events (HR 0.83; 95% CI, 0.74–0.94; *p* = 0.00). However, the benefits were primarily observed in non-dialysis patients, while the dialysis subgroup did not demonstrate significant results (HR 0.90; 95% CI, 0.75–1.08) [16]. The Kidney Disease: Improving Global Outcomes (KDIGO) guideline recommends against the initiation of statins in patients on dialysis due to the lack of significant cardiovascular benefit. However, due to limited data, the guideline recommends continuing statin therapy for patients already receiving the treatment [8]. Similarly, the American Heart Association/American College of Cardiology (AHA/ACC) guidelines identify CKD with an eGFR of 15–59 mL/min/1.73 m^2^ as a high-risk condition requiring consideration for statin initiation in primary prevention. This threshold indicates that statins may not offer cardiovascular benefits at an eGFR below 15 mL/min/1.73 m^2^ [9].

Patients on dialysis exhibit a unique cardiovascular phenotype [6,7,17,18]. The Cholesterol Treatment Trialists’ (CTT) collaboration analyzed 28 randomized trials and demonstrated that the effectiveness of statins in reducing major vascular events decreased progressively as eGFR declined [19]. The reduction effects of statins were 0.76 (99% CI, 0.70–0.81) for eGFR 45 to <60 mL/min/1.73 m^2^, 0.85 (99% CI, 0.75–0.96) for eGFR 30 to <45 mL/min/1.73 m^2^, and 0.85 (99% CI, 0.71–1.02) for eGFR < 30 mL/min/1.73 m^2^ in non-dialysis patients. For dialysis patients, the impact of statin therapy was not statistically significant, with a relative risk of 0.94 (99% CI, 0.79–1.11) [20]. In contrast, robust evidence exists on the effectiveness of statins in patients not on dialysis. A meta-analysis of 26 randomized trials reported a linear correlation, where a 1 mmol/L reduction in LDL-C at one year corresponded to a 22% decrease in major vascular events (95% CI, 20–24%; *p* < 0.0001) [21].

While randomized clinical trials provide a standard level of evidence, conducting these studies in dialysis patients is particularly challenging, given the relatively small sample size and strict inclusion criteria. As a result, real-world evidence becomes essential to capture the complexities of clinical practice and better understand treatment outcomes in this high-risk population [20]. This study aims to evaluate the safety and effectiveness of statins by including only patients receiving hemodialysis, providing real-world data to address the current insufficient evidence regarding the benefits and risks of statin therapy in this population [22].

## 2. Results

The study included 179 hemodialysis patients, of whom 115 were female (64.2%). The mean age of the participants was 66.6 ± 14.7 years, and the mean body mass index (BMI) was 28.6 ± 7.7 kg/m^2^. Among the participants, diabetes mellitus was present in 134 patients (74.9%), while 168 patients (93.9%) had hypertension. Cardiovascular events were observed in 95 patients (53.1%), which included acute coronary syndrome (ACS) in 19 patients (10.6%), heart failure (HF) in 25 patients (14%), and both ACS and HF in 49 patients (27.4%). Stroke was documented in 64 patients (35.8%), and peripheral vascular disease (PVD) was identified in 3 patients (1.7%). Regarding statin therapy, 148 patients (82.7%) were on statin therapy, with atorvastatin being the most frequently prescribed (69.3%), followed by simvastatin (7.8%) and rosuvastatin (5.6%). The distribution of sociodemographic and clinical variables is presented in Table 1. In Appendix A provides additional information regarding sociodemographic and clinical characteristics of patients stratified by individual statin types and non-statin therapy.

Table 2 outlines the baseline characteristics of patients stratified by statin users versus non-statin users. Compared to non-statin users, patients on statins were older, had higher BMI, and a greater burden of comorbidities—particularly diabetes (82.4% vs. 38.7%). Cardiovascular events were more common among statin users, with over half experiencing ACS, heart failure, or both, whereas most non-statin users had no such history. Prior stroke was also more frequent in statin users.

### 2.1. Lipid Profile Changes Among the Study Participants

Lipid profile changes were analyzed across four groups: participants receiving atorvastatin, rosuvastatin, simvastatin, and those not on statin therapy. The average duration between the first and third clinical visits was 380.8 days (SD = 105.6), which indicates that patients were typically followed for a mean of 12.7 months (Table 3).

At baseline, the non-statin group had the highest mean LDL-C level (2.678 mmol/L), followed by the rosuvastatin group (2.607 mmol/L), whereas the simvastatin group had the lowest (1.550 mmol/L). The atorvastatin group had an intermediate baseline LDL-C level (2.058 mmol/L).

By follow-up, LDL-C levels decreased in the atorvastatin and rosuvastatin groups while increased in the simvastatin and non-statin groups. Triglyceride levels declined in the atorvastatin and non-statin groups but increased in the rosuvastatin and simvastatin groups. Total cholesterol showed similar directional changes. HDL-C levels rose slightly in the atorvastatin and non-statin groups and declined in the rosuvastatin and simvastatin groups. Overall, lipid profile changes across all groups were not statistically significant (*p* > 0.05).

### 2.2. Association Between New Atherosclerotic Events and the Study Participants

The association between statin and non-statin therapy with atherosclerotic events was assessed using chi-square analysis (Table 4). Stroke was reported in the first, second, and third visits at 2.8%, 3.4%, and 5.0%, respectively. While the PVD with stroke incidence remained at 1.1% during all visits. No statistically significant association was found for atorvastatin (χ^2^ = 0.127, *p* = 0.72), simvastatin (χ^2^ = 1.389, *p* = 0.24), or the non-statin therapy group (χ^2^ = 1.29, *p* = 0.25). However, rosuvastatin use was significantly associated with a higher incidence of atherosclerotic events (χ^2^ = 6.48, *p* = 0.01). It is important to note that initiation of all statin therapies, including rosuvastatin, preceded the occurrence of the new atherosclerotic events. Also, in Appendix A, the mean survival times (in days) to atherosclerotic events by statin type.

Figure 1 displays the Kaplan–Meier survival curves comparing new atherosclerotic event-free survival across different statin groups and non-statin users. Overall, all groups demonstrated high survival probabilities, particularly within the first 300 days of follow-up. Patients on atorvastatin demonstrated the longest mean survival time, while those on rosuvastatin and simvastatin exhibited slightly earlier declines, although survival rates remained above 80% throughout the observation period. Non-statin users also maintained favorable survival, though a sharper decline was noted toward the end of follow-up. Despite these visual trends, Cox regression analysis did not show statistically significant differences between groups (*p* = 0.76), suggesting similar atherosclerotic event risk over time regardless of statin use (Table 5).

Cox proportional hazards regression was used to compare time to new atherosclerotic events between statin groups, using atorvastatin as the reference category. The overall model was not statistically significant (*p* = 0.76). None of the statin groups showed a significant difference in hazard compared to atorvastatin. Hazard ratios were 1.55 (95% CI: 0.19–12.78) for rosuvastatin, 1.06 (95% CI: 0.13–8.88) for simvastatin, and 1.94 (95% CI: 0.57–6.68) for the non-statin group. These wide confidence intervals reflect the small number of events (n = 14) and suggest that statin type was not a significant predictor of atherosclerotic outcomes in this cohort.

### 2.3. Comparison of Liver Function and Creatine Kinase Levels Across Groups

The comparison of liver function tests (LFTs) and creatine kinase (CK) levels among the study participants revealed no statistically significant changes in most parameters during the study period (*p* > 0.05) (Table 6). For the atorvastatin group, aspartate aminotransferase (AST) and alanine aminotransferase (ALT) levels increased slightly by the follow-up visit, while alkaline phosphatase (ALP) and CK levels showed minimal variation. Similarly, the rosuvastatin group exhibited slight increases in AST, ALT, and ALP levels, with a reduction in CK levels. In contrast, the simvastatin group showed decreasing trends in AST, ALT, and CK levels (*p* = 0.05). Among participants who did not receive statin therapy, minimal fluctuations were observed.

## 3. Discussion

To the best of our knowledge, this is the first real-world study to assess the outcomes associated with the use of statin therapy in patients with ESRD undergoing hemodialysis in Saudi Arabia. The study evaluated the efficacy and safety of statin therapy, focusing on changes from baseline throughout the follow-up period in lipid profile, in addition to monitoring the development of atherosclerotic events and adverse events related to statin use. Although statin use—primarily atorvastatin—was common in the cohort, the modest reductions in LDL-C did not result in significant protection against atherosclerotic events. Most patients initially remained unaffected; however, a gradual progression of atherosclerotic events is evident over time. These findings are consistent with evidence from the AURORA and 4D trials. Statins have been linked to moderate reductions in LDL-C levels in HD patients but have demonstrated no significant effect on mortality or CV outcomes [15,22]. The study finding suggests that the clinical utility of statins in Saudi hemodialysis populations does not translate into clinical outcomes.

At baseline in the current study, a high rate of CV events, such as ACS and stroke, was noted more commonly among the atorvastatin group, who had a diagnosis of diabetes. Lipid profile alterations were examined in four groups in our study: those not on statin medication, and those using atorvastatin, rosuvastatin, and simvastatin. All groups’ changes in lipid profiles were not observed overall (*p* > 0.05). These results are in line with the outcomes of the 4D trial [22].

Atorvastatin (20 mg daily) was assessed in 1255 HD patients with type 2 diabetes. After four weeks of atorvastatin treatment, LDL-C decreased by 42%; however, the key composite endpoint did not significantly decrease. These results imply that even though statins may theoretically improve lipid profiles in hemodialysis patients, our study supports important trials by showing that alterations in lipid parameters might not be as noticeable in this population as anticipated, which reflects the distinct lipid metabolism features of hemodialysis-treated patients.

The safety profile of statins in our cohort analysis revealed a slight increase in AST and ALT, although values remained within the normal range. Additionally, our study indicates modest fluctuations in CK levels among participants receiving statins. However, these changes lacked statistical significance and clinical relevance. While our findings support the safe use of statins in hemodialysis patients, this analysis does not imply a reduced need for regular monitoring of liver enzymes per clinical guidelines. This corresponds with the SHARP and AURORA studies, which underscored the significance of monitoring hepatic function during statin therapy while noting a low incidence of adverse effects. Moreover, CK levels exhibited no substantial fluctuations between visits, indicating a minimal risk of muscle-related adverse effects.

In our study, rosuvastatin use was associated with a higher incidence of atherosclerotic events (χ^2^ = 6.48, *p* = 0.01). However, given the small sample size of participants on rosuvastatin (n = 10), this finding should be interpreted with caution. While the statistical association suggests a potential link, the limited number of patients reduces the generalizability of the result and raises the possibility of random variation. Larger cohort analysis in Saudi Arabia is necessary to confirm these preliminary observations of rosuvastatin use and to guide clinical decision-making for patients undergoing statin therapy and hemodialysis. Meanwhile, data from the AURORA study did not show an increase in cardiovascular events or mortality associated with statin use in hemodialysis patients. Additionally, the 4D study, which evaluated the impact of atorvastatin on diabetic patients undergoing dialysis, demonstrated no statistically significant effect on cardiovascular outcomes.

Our study aligns with previous clinical trials that have highlighted the distinct pathophysiology of cardiovascular disease in dialysis patients. Statins therapy alone may not be sufficient to address the significant involvement of other variables, including inflammation, arterial stiffness, and other atherogenic processes. This reinforces the notion that cardiovascular risk in hemodialysis patients extends beyond dyslipidemia management. Nevertheless, our research has several strengths and limitations. One of the limitations is that it did not assess the impact of statins on other lipid parameters that are more closely associated with cardiovascular risk, such as Lp(a). Statins have been shown to slightly increase Lp(a) levels; however, this change has not demonstrated clinical relevance [12]. Additionally, concurrent medications like antihypertensives and antidiabetics were not accounted for in our analysis, which may independently influence cardiovascular outcomes. Furthermore, the observational study design and reliance on existing patient records introduce additional limitations, potentially restricting the ability to establish causal relationships between statin use and clinical outcomes. Despite this, our study has several strengths reflecting its clinical importance and research value. First, it provides essential real-world data on statin use and its efficacy and safety outcomes in routine clinical practice, ensuring the applicability of the findings. Second, the study targets a high-risk population of ESRD patients on HD with dyslipidemia, which is usually an underrepresented population, covering a research gap in which randomized clinical trials showed limited statins benefit. Although the sample size is relatively small, given the nature of the disease and the high-risk population, the sample size remains appropriate for exploratory analysis. Moreover, monitoring of safety parameters associated with statin use detailing the effect of each statin type provided additional insights into the study.

## 4. Materials and Methods

### 4.1. Study Design

This retrospective cohort study examined the effects of statin use in adult ESRD patients (≥18 years) on hemodialysis with documented dyslipidemia admitted between 1 January 2018 and 30 December 2019. Inclusion criteria required each patient to have at least two lipid profile measurements within the study period, with the first measurement representing the baseline and subsequent measurement(s) as follow-up. Patients were excluded if they had chronic kidney disease (CKD), were on peritoneal dialysis, were younger than 18 years, or had less than two lipid profile measurements. In addition, patients with incomplete lipid data or missing clinical outcomes were excluded from the analysis.

### 4.2. Study Endpoints

The primary endpoints evaluated the clinical outcomes of statins by assessing changes in specific lipid parameters, including low-density lipoprotein cholesterol (LDL-C), high-density lipoprotein cholesterol (HDL-C), triglycerides (TG), total cholesterol (TC), and the incidence of atherosclerotic events (defined as incidence of new event of stroke, or Peripheral Artery/vascular Disease (PAD/PVD), or stroke and PVD). The secondary endpoint assessed the safety of statin use through monitoring of liver enzymes, including alkaline phosphatase (ALP), alanine transaminase (ALT), and aspartate aminotransferase (AST). Additionally, creatine kinase (CK) levels, myopathy, and rhabdomyolysis were monitored.

### 4.3. Statistical Analysis

Quantitative variables are presented as mean ± standard deviation (SD), along with minimum and maximum values where applicable. Categorical variables are expressed as frequencies and percentages. Between-group comparisons for categorical variables (e.g., demographics, clinical characteristics) were conducted using the chi-square test. Continuous variables (e.g., lab values) were compared between statin users and non-users using independent-sample *t*-tests. Survival analysis was performed to evaluate the time to new atherosclerotic events among different statin groups and non-statin users. Kaplan–Meier survival curves were generated, and differences in survival distributions were visually assessed. The log-rank test was applied to compare event-free survival across statin types. Cox proportional hazards regression analysis was used to examine the association between statin type and the risk of new atherosclerotic events, using atorvastatin as the reference category. Hazard ratios (HR) and 95% confidence intervals (CIs) were reported. The model’s overall significance was evaluated using the Wald chi-square statistic. All statistical analyses were performed using IBM SPSS Statistics software (version 27.0; SPSS Inc., Chicago, IL, USA). A *p*-value of <0.05 was considered statistically significant.

## 5. Conclusions

In conclusion, this study evaluated the efficacy and safety of statin therapy in hemodialysis patients with dyslipidemia. Although modest reductions in LDL-C levels were observed in the atorvastatin and rosuvastatin groups, statin therapy did not significantly reduce the incidence of atherosclerotic events. These findings suggest that routine statin use in hemodialysis patients may offer limited cardiovascular benefit and underscore the need for individualized treatment strategies that account for the complex pathophysiology of cardiovascular disease in this population. Importantly, no clinically or statistically significant elevations in liver enzymes were observed, supporting the overall safety of statin therapy in this cohort.

## Figures and Tables

**Figure 1 pharmaceuticals-18-00911-f001:**
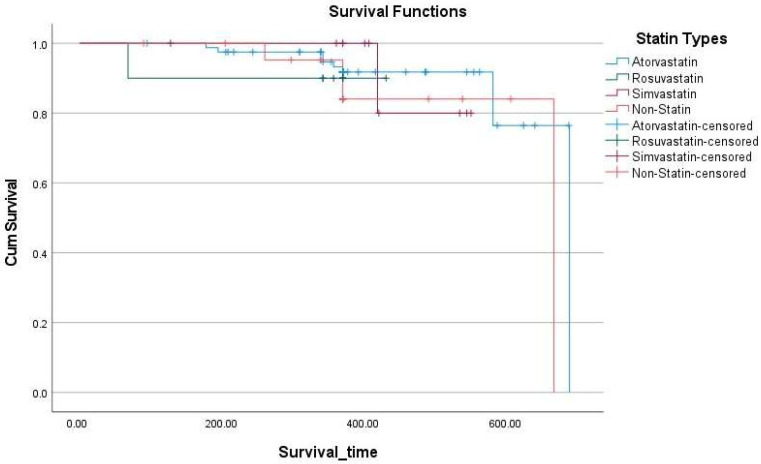
Kaplan–Meier survival curves comparing time to new atherosclerotic events by statin type.

**Table 1 pharmaceuticals-18-00911-t001:** Distribution of sociodemographic and clinical variables among patients (n = 179).

Socio-Economic and Clinical Variables	No. of Patients (n)	Percentage (%)
Male	64	35.8
Female	115	64.2
Age (Mean ± SD)	66.6 ± 14.7 years	
Body Mass Index (Mean ± SD)	28.6 ± 7.7 kg/m^2^	
Comorbidities		
Diabetes Mellitus	134	74.9
Hypertension	168	93.9
CV events		
None	84	46.9
ACS	19	10.6
HF	25	14.0
ACS and HF	49	27.4
RHD with MS	2	1.1
Atherosclerotic Events History		
None	108	60.3
Stroke	64	35.8
PAD	3	1.7
Stroke and PAD	4	2.2
Statin Therapy		
Atorvastatin	124	69.3
Rosuvastatin	10	5.6
Simvastatin	14	7.8
Not on Statin Therapy	31	17.3
Distribution of Atorvastatin Doses		
10 mg	15	12.1
20 mg	45	36.3
40 mg	56	45.2
60 mg	2	1.6
80 mg	6	4.8
Distribution of Rosuvastatin Doses		
10 mg	5	50
20 mg	3	30
40 mg	2	20
Distribution of Simvastatin Doses		
10 mg	3	21.4
20 mg	8	57.2
40 mg	3	21.4

ACS: Acute coronary syndrome; CV: Cardiovascular events; HF: Heart failure; PAD: Peripheral artery disease; RHD with MS: Rheumatic heart disease with mitral stenosis.

**Table 2 pharmaceuticals-18-00911-t002:** Sociodemographic and clinical characteristics of patients stratified by statin users versus non-statin users.

Socio-Economic and Clinical Variables	Statin Users n (%)	Non-Statin Users n (%)
Male	49 (33.1%)	15 (48.4%)
Female	99 (66.9%)	16 (51.6%)
Age (Mean ± SD)	67.62 (13.23)	61.90 (19.91)
Body Mass Index (Mean ± SD)	29.40 (7.79)	25.06 (6.17)
Comorbidities		
Diabetes Mellitus	122 (82.4%)	12 (38.7%)
Hypertension	140 (94.6%)	28 (90.3%)
CV events		
None	60 (40.5%)	24 (77.4%)
ACS	18 (12.2%)	1 (3.2%)
HF	22 (14.9%)	3 (9.7%)
ACS and HF	46 (31.1%)	3 (9.7%)
RHD with MS	2 (1.4%)	0 (0.0%)
Atherosclerotic Events History		
None	86 (58.1%)	22 (71.0%)
Stroke	55 (37.2%)	9 (29.0%)
PAD	3 (2.0%)	0 (0.0%)
Stroke and PAD	4 (2.7%)	0 (0.0%)

ACS: Acute coronary syndrome; CV: Cardiovascular events; HF: Heart failure; PAD: Peripheral artery disease; RHD with MS: Rheumatic heart disease with mitral stenosis.

**Table 3 pharmaceuticals-18-00911-t003:** Comparison of lipid profile changes among the participant groups.

Participant Groups	Lipid Profile (mmol/L)	Duration	Mean	SD	t-Value	*p*-Value
Atorvastatin	LDL-C	Baseline	2.058	0.818	0.609	0.54
		Follow-up	2.003	0.769		
	TG	Baseline	1.730	1.113	0.642	0.52
		Follow-up	1.656	1.019		
	TC	Baseline	3.769	1.059	1.062	0.29
		Follow-up	3.654	0.974		
	HDL-C	Baseline	0.900	0.245	−0.969	0.33
		Follow-up	0.925	0.262		
Rosuvastatin	LDL-C	Baseline	2.607	0.942	1.160	0.28
		Follow-up	2.113	1.136		
	TG	Baseline	1.482	0.855	−0.723	0.49
		Follow-up	1.766	1.003		
	TC	Baseline	4.398	1.165	1.143	0.29
		Follow-up	3.785	1.390		
	HDL-C	Baseline	1.011	0.304	0.869	0.41
		Follow-up	0.930	0.344		
Simvastatin	LDL-C	Baseline	1.550	0.520	−1.608	0.15
		Follow-up	1.901	0.651		
	TG	Baseline	1.480	0.865	−1.555	0.16
		Follow-up	1.845	0.963		
	TC	Baseline	3.268	0.818	−1.188	0.27
		Follow-up	3.632	0.863		
	HDL-C	Baseline	0.997	0.333	0.570	0.58
		Follow-up	0.952	0.228		
Not on Statin Therapy	LDL-C	Baseline	2.678	0.630	−0.796	0.44
		Follow-up	2.820	0.821		
	TG	Baseline	1.351	0.527	0.553	0.59
		Follow-up	1.282	0.458		
	TC	Baseline	4.391	0.670	−0.586	0.56
		Follow-up	4.515	0.915		
	HDL-C	Baseline	1.089	0.269	−0.495	0.63
		Follow-up	1.107	0.235		

**Table 4 pharmaceuticals-18-00911-t004:** Association between participant groups and new atherosclerotic events.

Participant Groups		Atherosclerotic Events		Total (%)	Chi-Square Test χ^2^	*p*-Value
		No (%)	Yes (%)			
Atorvastatin	No	51 (28.5)	4 (2.2)	55 (30.7)	0.127	0.72
	Yes	113 (63.1)	11 (6.1)	124 (69.3)		
Rosuvastatin	No	157 (87.7)	12 (6.7)	169 (94.4%)	6.48	0.01
	Yes	7 (3.9)	3 (1.7)	10 (5.6)		
Simvastatin	No	150 (83.8)	15 (8.4)	165 (92.2)	1.389	0.24
	Yes	14 (7.8)	0 (0.0)	14 (7.8)		
Not on Statin Therapy	No	134 (74.9)	14 (7.8)	148 (82.7)	1.29	0.25
	Yes	30 (16.8)	1 (0.6)	31 (17.3)		

**Table 5 pharmaceuticals-18-00911-t005:** Cox proportional hazards regression comparing new atherosclerotic event risk by statin type.

	B	SE	Wald	df	Sig.	Exp(B)	95.0% CI for Exp(B)	
							Lower	Upper
Statin Types			1.186	3	0.756			
Rosuvastatin	0.436	1.077	0.164	1	0.685	1.547	0.187	12.778
Simvastatin	0.059	1.084	0.003	1	0.956	1.061	0.127	8.882
Non-Statin	0.665	0.630	1.114	1	0.291	1.944	0.566	6.676

Atorvastatin is the reference category for the categorical variable Statin Types. The overall row labeled Statin Types represents the joint significance test (Wald χ^2^) for the variable. Hazard ratios [Exp(B)] represent the relative hazard compared to Atorvastatin. CI = Confidence Interval; SE = Standard Error; Sig. = Significance (*p*-value), and B = the regression coefficient.

**Table 6 pharmaceuticals-18-00911-t006:** Comparison of liver function tests (LFTs) and creatine kinase (CK) among the participant groups.

Participant Groups	LFTs (U/L)	Duration	Mean	SD	t-Value	*p*-Value
Atorvastatin	AST	Baseline	17.57	9.56	−0.627	0.53
		Follow-up	24.08	97.09		
	ALT	Baseline	14.27	8.42	−0.945	0.35
		Follow-up	24.08	97.09		
	ALP	Baseline	147.61	97.83	−1.766	0.08
		Follow-up	160.57	100.41		
	CK Level	Baseline	56.41	46.04	0.454	0.65
		Follow-up	53.48	53.37		
Rosuvastatin	AST	Baseline	14.22	5.24	−0.388	0.71
		Follow-up	15.44	8.05		
	ALT	Baseline	12.78	5.45	−0.944	0.37
		Follow-up	15.44	8.05		
	ALP	Baseline	133.11	36.87	−0.282	0.78
		Follow-up	145.00	131.26		
	CK Level	Baseline	97.22	74.04	1.870	0.10
		Follow-up	46.44	23.51		
Simvastatin	AST	Baseline	23.56	24.22	1.445	0.19
		Follow-up	11.22	3.87		
	ALT	Baseline	18.44	21.89	0.943	0.37
		Follow-up	11.22	3.87		
	ALP	Baseline	122.89	58.88	−0.167	0.87
		Follow-up	126.56	20.09		
	CK Level	Baseline	56.50	35.61	2.353	0.05
		Follow-up	37.63	16.50		
Not on Statin Therapy	AST	Baseline	16.68	8.13	0.620	0.54
		Follow-up	14.89	15.82		
	ALT	Baseline	13.11	9.83	−0.987	0.34
		Follow-up	14.89	15.82		
	ALP	Baseline	180.37	295.29	0.412	0.69
		Follow-up	168.11	186.16		
	CK Level	Baseline	46.63	28.92	−0.760	0.46
		Follow-up	57.95	67.48		

## Data Availability

The data are secured to guarantee patients’ privacy.

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
