# Peer review of "Evaluation of Statins Use in Hemodialysis Patients: A Retrospective Analysis of Clinical and Safety Outcomes"

_pharmaceuticals, 2025, doi:10.3390/ph18060911_

Round 1
Reviewer 1 Report
Comments and Suggestions for Authors
I read with interest the paper titled "Evaluation of Statins Use in Hemodialysis Patients: A Retrospective Analysis of Clinical and Safety Outcomes"
I have minor comments that could enhance the manuscript:
1. Please refer the duration of the retrospective study within the abstract.
2. Table 1 present the doses of Atorvastatin; how about the other statins?
3. Table 3 is quite doubious. Rosuvastatin was associated with atherosclerotic events but the causation cannot be assessed? Are those statins introduced right after the event or the statins intake started before the CV event?
4. In addition to those analysis involving statins only, the other drugs of each patients should be taken in account to draw conclusions. Most of those patients are polimedicated with hypertension and diabetes (and other diseases).
5. Since the population is from hemodialysis patients, I expect more discussion around that. Results, as they are presented could be for any sample. What's the uniqueness of the special population that is present in the study?
Author Response
We thank the reviewer for their valuable insights and constructive comments. We have revised the manuscript accordingly. Below, we provide a detailed response to each point raised, along with the corresponding changes that have been made.
Comment 1:
"Please refer the duration of the retrospective study within the abstract".
Response 1:
We thank the reviewer for highlighting this important point. We have updated the abstract in the method section accordingly.
Comment 2:
"Table 1 presents the doses of Atorvastatin; how about the other statins?"
Response 2:
We thank the reviewer for the comment. We have updated table-1 to include the full distribution of doses for Rosuvastatin and Simvastatin, in addition to Atorvastatin. The revised table now clearly presents the dosage breakdown across all statins reported in the study.
Comment 3:
"Table 3 is quite doubious. Rosuvastatin was associated with atherosclerotic events but the causation cannot be assessed? Are those statins introduced right after the event or the statins intake started before the CV event?"
Response 3:
Comment 4:
"In addition to those analysis involving statins only, the other drugs of each patients should be taken in account to draw conclusions. Most of those patients are polimedicated with hypertension and diabetes (and other diseases). "
Response 4:
"Since the population is from hemodialysis patients, I expect more discussion around that. Results, as they are presented could be for any sample. What's the uniqueness of the special population that is present in the study?"
Response 5:
We thank the reviewer for the comment. More emphasis was made and detailed in the revised manuscript.Reviewer 2 Report
Comments and Suggestions for Authors
Abdulmalik and coworkers presented an interesting study: Evaluation of statins use in hemodialysis patients: a retrospective analysis of clinical and safety outcomes.
The authors addressed an interesting topic, included 179 HD patients, used an appropriate methodology, and presented results and conclusions.
However, the study does not provide information on its statistical power and lacks details about the novelty and contribution of its findings to the existing body of knowledge.
Author Response
We thank the reviewer for the valuable insights and constructive comments. We have revised the manuscript accordingly. Below, we provide a detailed response to each point raised, along with the corresponding changes that have been made.
Comment 1:
"the study does not provide information on its statistical power".
Response 1:
Comment 2:
" lacks details about the novelty and contribution of its findings to the existing body of knowledge."
Response 2:
We thank the reviewer for the comment. More elaborations on the study’s strengths have been added and highlighted in the discussion.
Reviewer 3 Report
Comments and Suggestions for Authors
The study by Alotaibi et al is a retrospective study to analyze the use of statins among hemodialysis patients. Previous large interventional studies among dialysis patients did not find a benefit regarding CV event by statins, in contrast to CKD patients. This retrospective study is in line with these previous large interventional studies. While the present study is a retrospective and small study – especially for the subgroups rosuvastatin and simvastatin – it is important to understand such potential associations in different cohorts – such as Saudi Arabia. Therefore, this study is of potential interest.
Here are some additional comments:
- Abstract: “Observational studies indicate that statins may improve survival rates in hemodialysis patients.” Many interventional studies did not find improved survival by statin use among HD patients. Therefore, such a statement at the beginning could be misleading.
- Abstract: “LDL-C levels showed modest changes across all groups.” Is it changes or differences between the statin vs non-statin group?
- Abstract: “Among non-statin users, LDL-C and total cholesterol increased, triglycerides decreased, and HDL-C rose slightly.” The LDL values should also be provided as for the other groups.
- Abstract: “While statins reduced LDL-C levels, they did not significantly prevent new atherosclerotic events.” This conclusion may also be misleading as atorvastatin did not decrease LDL levels in a relevant amount and simvastatin rather increased LDL-C. Moreover, the results section of the abstract does not mention atherosclerotic events.
- Introduction: “Additional pathophysiological factors, such as arterial stiffness and calcification, significantly influence ESRD patients' cardiovascular [7,13].” The sentence is not complete.
- Table 1. To understand any potential differences in the statin / non-statin groups, the table could stratify the patients according to statins.
- Results section 2.1. The information from the table is duplicated in the text. It is not needed to repeat the numbers. It would be sufficient to state that there are no significant differences between baseline and follow-up. Moreover, it would be interesting to add the information whether the different groups differ in the different parameters, e.g. LDL-C between the four groups at baseline and at follow-up.
- It is not clear how long the follow-up was for the individual patients. A Kaplan-Meier and multivariate Cox analysis could be performed to compare the different groups regarding CV events.
- Discussion: “Although statin use—primarily atorvastatin—was common in the cohort, reductions in LDL-C did not result in significant protection against atherosclerotic events” – As there was no significant reduction in the LDL-C levels, this should be reworded.
- Discussion: “These results are in line with the outcomes of the 4D trial.” Citation is missing.
Author Response
We thank the reviewer for the valuable insights and constructive comments. We have revised the manuscript accordingly. Below, we provide a detailed response to each point raised, along with the corresponding changes that have been made.
Comment 1:
"Abstract: “Observational studies indicate that statins may improve survival rates in hemodialysis patients.” Many interventional studies did not find improved survival by statin use among HD patients. Therefore, such a statement at the beginning could be misleading."
Response 1:
We thank the reviewer for the valuable comment. We agree that the original statement may have overstated the observational evidence and did not adequately reflect findings from RCTs. Accordingly, we have revised the abstract. The updated sentence reads:
“Although large randomized controlled trials have generally not demonstrated a survival benefit associated with statin therapy among patients receiving hemodialysis, limited observational studies have reported potential associations with improved clinical outcomes in this population.”
Comment 2:
"Abstract: “LDL-C levels showed modest changes across all groups.” Is it changes or differences between the statin vs non-statin group?"
Response 2:
We thank the reviewer for this valuable comment. The original statement referred to within-group changes rather than between-group differences. We have revised the sentence in the abstract to:
“Modest reductions in LDL-C levels were observed in the rosuvastatin and atorvastatin groups, whereas slight increases were noted in the simvastatin and non-statin groups. None of these within-group changes were statistically significant.”
Comment 3:
"Abstract: “Among non-statin users, LDL-C and total cholesterol increased, triglycerides decreased, and HDL-C rose slightly.” The LDL values should also be provided as for the other groups."
Response 3:
We thank the reviewer for the comment, we have updated the abstract in the result section accordingly. comment 4:"Abstract: “While statins reduced LDL-C levels, they did not significantly prevent new atherosclerotic events.” This conclusion may also be misleading as atorvastatin did not decrease LDL levels in a relevant amount and simvastatin rather increased LDL-C. Moreover, the results section of the abstract does not mention atherosclerotic events."
Response 4:
We thank the reviewer for the comment. To more accurately reflect the results, we have revised the abstract conclusion to:
“Although modest reductions in LDL-C levels were observed in the atorvastatin and rosuvastatin groups, statin therapy did not reduce the incidence of atherosclerotic events in hemodialysis patients with dyslipidemia.”
Comment 5:
"Introduction: “Additional pathophysiological factors, such as arterial stiffness and calcification, significantly influence ESRD patients' cardiovascular [7,13].” The sentence is not complete."
Response 5:
Comment 6:
"Table 1. To understand any potential differences in the statin / non-statin groups, the table could stratify the patients according to statins."
Response 6:
We appreciate the reviewer’s insightful suggestion. In response, we have revised the presentation of baseline characteristics to enhance clarity. Specifically, we now include Table 1a:
- Table 1a summarizes the same characteristics grouped by overall statin use versus non-use, providing a clearer high-level view of group differences. This presentation ensures both clinical specificity and interpretive clarity, directly supporting the comparative analyses that follow in the manuscript.
Comment 7:
"Results section 2.1. The information from the table is duplicated in the text. It is not needed to repeat the numbers. It would be sufficient to state that there are no significant differences between baseline and follow-up. Moreover, it would be interesting to add the information whether the different groups differ in the different parameters, e.g. LDL-C between the four groups at baseline and at follow-up."
Response 7:
We thank the reviewer for the valuable comment. We have revised Section 2.1 to eliminate redundant numerical values already presented in Table 2. The text now summarizes lipid profile trends using concise directional language and includes selective baseline LDL-C values to highlight inter-group differences. Follow-up data are described narratively without repeating table values.Comment 8:
"It is not clear how long the follow-up was for the individual patients. A Kaplan-Meier and multivariate Cox analysis could be performed to compare the different groups regarding CV events."
Response 8:
We appreciate the reviewer’s request for clarification regarding patient follow-up time. Based on the full dataset, the average duration between the first and third clinical visits was 380.8 days (SD = 105.6). This indicates that patients were typically followed for a mean of 12.7 months, with variability captured by the standard deviation (Added in the result section).We also confirm that time-to-event analyses (Kaplan-Meier and Cox regression) accounted for censoring, ensuring appropriate handling of varying follow-up durations across patients. We clarify that cardiovascular events were documented at baseline, whereas incident events occurring during follow-up were recorded as atherosclerotic events. In response, we performed both Kaplan-Meier survival analysis and multivariate Cox regression to compare time to atherosclerotic events across different statin groups. These analyses have been added to the Results section.
Comment 9:
"Discussion: “Although statin use—primarily atorvastatin—was common in the cohort, reductions in LDL-C did not result in significant protection against atherosclerotic events” – As there was no significant reduction in the LDL-C levels, this should be reworded."
Response 9:
We thank the reviewer for the comment. The sentence has been revised.
“Although statin use—primarily atorvastatin—was common in the cohort, the modest reductions in LDL-C did not result in significant protection against atherosclerotic events, where most patients remained unaffected initially, a gradual progression of atherosclerotic events is evident over time.”
Comment 10:
"Discussion: “These results are in line with the outcomes of the 4D trial.” Citation is missing."
Response 10:
Thank you for this point. We added the citation.Round 2
Reviewer 2 Report
Comments and Suggestions for Authors
The authors correctly accepted the comments and improved the manuscript.
Reviewer 3 Report
Comments and Suggestions for Authors
No further comments.